# Protocol for the development of SPIRIT and CONSORT extensions for randomised controlled trials with surrogate primary endpoints: SPIRIT-SURROGATE and CONSORT-SURROGATE

Anthony Muchai Manyara [1], Philippa Davies,[2] Derek Stewart,[3] Christopher J Weir [4], Amber Young [2], Nancy J Butcher [5,6] Sylwia Bujkiewicz,[7] An-Wen Chan,[8,9] Gary S Collins [10], Dalia Dawoud,[11] Martin Offringa [6], Mario Ouwens,[12] Joseph S Ross,[13,14] Rod S Taylor,[1,15] Oriana Ciani[16]

We wish to acknowledge our dear colleague Professor Amber Young who lost her brave battle with cancer and passed away on 17th September 2022.

For numbered affiliations see end of article.

**Correspondence to**
Dr Anthony Muchai Manyara; anthony.manyara@glasgow.ac.uk

## ABSTRACT

**Introduction** Randomised controlled trials (RCTs) may use surrogate endpoints as substitutes and predictors of patient-relevant/participant-relevant final outcomes (eg, survival, health-related quality of life). Translation of effects measured on a surrogate endpoint into health benefits for patients/participants is dependent on the validity of the surrogate; hence, more accurate and transparent reporting on surrogate endpoints is needed to limit misleading interpretation of trial findings. However, there is currently no explicit guidance for the reporting of such trials. Therefore, we aim to develop extensions to the SPIRIT (Standard Protocol Items: Recommendations for Interventional Trials) and CONSORT (Consolidated Standards of Reporting Trials) reporting guidelines to improve the design and completeness of reporting of RCTs and their protocols using a surrogate endpoint as a primary outcome.

**Methods and analysis** The project will have four phases: phase 1 (literature reviews) to identify candidate reporting items to be rated in a Delphi study; phase 2 (Delphi study) to rate the importance of items identified in phase 1 and receive suggestions for additional items; phase 3 (consensus meeting) to agree on final set of items for inclusion in the extensions and phase 4 (knowledge translation) to engage stakeholders and disseminate the project outputs through various strategies including peer-reviewed publications. Patient and public involvement will be embedded into all project phases.

**Ethics and dissemination** The study has received ethical approval from the University of Glasgow College of Medical, Veterinary and Life Sciences Ethics Committee (project no: 200210051). The findings will be published in open-access peer-reviewed publications and presented in conferences, meetings and relevant forums.

## STRENGTHS AND LIMITATIONS OF THIS STUDY

⇒ We will follow the EQUATOR (Enhancing the QUAlity and Transparency Of health Research) Network's recommended steps for developing a health research reporting guideline.
⇒ The Delphi study will target an international and multidisciplinary group of participants.
⇒ Patient and public involvement will be integrated in all phases of the study.
⇒ Although we will target international participation, Delphi study and consensus meeting will be conducted in English which could limit participation from non-English speaking settings.

## INTRODUCTION

Evidence for the effectiveness of interventions should ideally come from randomised controlled trials (RCTs)[1] [2] that assess a patient/study participant relevant final outcome (PRFO): a measurement that reflects how an individual feels, functions or survives,[3] such as mortality or health-related quality of life. Nevertheless, in order to meet the scientific, ethical and regulatory requirements, the conduct and delivery of many RCTs have become increasingly resource and time intensive.[4] By reducing follow-up time, sample size and cost, a surrogate endpoint that substitutes for a PRFO can provide a potentially attractive solution for improving trial efficiency.[5] This efficiency allows for early detection of intervention effects[6] which could lead to accelerated approval of interventions prior to confirmation of benefit on

the PRFO[7] or when there is lack of effect, stopping of trials or roll out of interventions with no health benefit.

Over the last decade, a number of biomarkers (an objectively measured molecular, histological, radiographic or physiological characteristic used as an indicator of response to an intervention)[5] have been accepted as surrogate endpoints in the regulatory approval of pharmaceuticals and medical devices, for example, blood pressure and low density lipoprotein-cholesterol within the cardiovascular disease context.[8] However, for a wider range of interventions including surgical, organisational and public health interventions, the so-called intermediate outcomes (ie, an outcome that can be measured earlier than an effect on PRFO and is predictive of the intervention effect on the PRFO)[3] have been used with the aim of replacing and predicting for a PRFO (e.g., hospice enrolment instead of mortality with an intervention aimed at improving end of life care;[9] fruit and vegetable consumption instead of cardiovascular events for a behavioural intervention designed to improve cardiovascular risk[10]). To be accepted as a valid surrogate endpoint, a biomarker or intermediate outcome needs to both: (1) reliably predict the PRFO in individual participants ('individual-level' or 'patient-level' surrogacy) and (2) the intervention effect on surrogate endpoint should reliably predict the intervention effect on the PRFO ('trial-level' surrogacy) which is assessed through a meta-analytic model of data on both outcomes typically obtained from RCTs.[11 12] In summary, surrogacy validation depends on the use of various statistical methods, including meta-analysis of trial aggregate data and/or individual patient data,[13 14] principal stratification,[15] causal inference,[16 17] information theory[18] and/or bivariate network meta-analysis methods.[19 20]

Despite their potential benefits, the use of surrogate endpoints in health and policy decision making remains controversial. Some regulatory approvals based on a surrogate endpoint have led to the use of interventions that have resulted in more harm than health benefit for patients due to the surrogate not being in the pathway of the PRFO or unintended effects of the intervention.[21–24] Additionally, RCTs using a surrogate endpoint primary outcome have been shown to overestimate treatment effects (adjusted ORs: 1.46, 95% CI: 1.05–2.04) compared with RCTs with PRFO as a primary outcome.[25] It is therefore important that RCTs using surrogate endpoints are appropriately designed and reported, for example, explicitly state that the primary outcome is a surrogate endpoint, give a clear rationale for its use and provide evidence of its validation (ie, prior evidence of the association between the intervention effect on the putative surrogate endpoint and PRFO).[7] However, a retrospective analysis of 626 RCTs published in 2005 and 2006 found that 109 (17%) used a surrogate primary endpoint, and of these, only a third discussed whether the surrogate endpoint was a valid predictor of health benefit on a PRFO.[26] Similarly, a more recent review of 220 cardiovascular intervention trials using surrogate biomarkers found that only 59 (27%) had confirmatory evidence validating the benefits of interventions on a PRFO.[27]

Reporting guidelines can improve transparency and completeness in the reporting of RCTs at both the protocol and final report stages. The two prominent trial guidelines are: SPIRIT (Standard Protocol Items: Recommendations for Interventional Trials) 2013 statement is a 33-item checklist used to report RCT protocols;[28] and CONSORT (Consolidated Standards of Reporting Trials) 2010 statement is a 25-item checklist, which has improved reporting of completed trials.[29 30] Although SPIRIT and CONSORT and related extensions, including the SPIRIT-PRO,[31] CONSORT-PRO[32] and the ongoing CONSORT-Outcomes,[33] provide general guidance on trial outcome reporting, there remains no standard evidence-based guideline for the reporting of RCTs with a surrogate primary outcome. Therefore, in this study, we aim to develop specific extensions to report RCT protocols and final trial reports that use a surrogate primary endpoint: SPIRIT-SURROGATE and CONSORT-SURROGATE, respectively. These extensions will improve the transparency of reporting and design of RCTs with a surrogate endpoint as a primary outcome. Such improved reporting should enable the evidence base for surrogate endpoints to be more effectively scrutinised and used for interpretation of trial findings by patients, clinicians and healthcare policy makers. This protocol describes the methods that will be used in developing these extensions.

## METHODS

Our methodology will be guided by: the EQUATOR (Enhancing the QUAlity and Transparency Of health Research) Network's recommended steps for developing a health research reporting guideline,[34] and methodological considerations used to develop other recent or ongoing SPIRIT/CONSORT extensions (e.g., ACE (Adaptive designs CONSORT Extension),[35] CONSORT-ROUTINE[36] and dose-finding CONSORT extension[37])

The project will be overseen by a project management group (PMG) and an advisory executive committee (EC). The PMG includes the lead investigators (OC and RT) and project manager (AMM) responsible for the day-to-day management of the project plus project co-investigators (PD, AY and CW) and patient and public involvement (PPI) lead (DS). The EC is an international and multidisciplinary group (JSR (chair), NJB, SB, AWC, GSC, DD, MO and Mario Ouwens) providing strategic oversight of the project and will contribute to the dissemination, endorsement, and implementation of developed extensions.

Figure 1 shows the project phases, timelines, activities in each phase and the integrated knowledge translation and PPI which are described in more detail below.

### Phase 1: literature reviews
A detailed protocol for this phase has been prepared, prospectively registered[38] and submitted for publication.

**Figure 1** Project phases, timelines, activities in each phase (middle), with integrated knowledge translation (left) and patient and public involvement (right). Timelines include preparatory work before start of each phase. Adapted from Kwakkenbos *et al*[62]. RCTs, Randomised Controlled Trials.

Briefly, we will undertake two separate literature reviews that include a scoping review and a 'targeted review'. The scoping review, conducted using the Arksey and O'Malley six-stage methodological framework,[39] will seek to explore the current understanding, limitations, acceptability and guidance on using surrogate endpoints in RCTs and generate two outputs: candidate items ('long-list') for rating through a Delphi exercise and a contact list of surrogate content experts who will be invited to participate in the exercise. Furthermore, one of the issues to be explored in the scoping review will be an appropriate and comprehensive working definition of surrogate endpoint, starting from the most commonly reported definition in 2001 by the National Institutes of Health (ie, a biomarker or laboratory measure intended to substitute and predict for a variable that reflects how patients feel, function or how long they survive).[5] The targeted review will seek to systematically identify recent protocols and full reports of RCTs that have used surrogate endpoints as primary outcomes. It will serve two purposes: (1) identify researchers who have used surrogate endpoints who will be invited to participate in the Delphi exercise and (2) the identified protocols and reports will inform a detailed contemporary analysis of the completeness of reporting of RCTs with surrogate endpoint primary outcomes.

## Phase 2: Delphi study

The primary objective of this phase will be to rate candidate items generated in phase 1. Secondary objectives will be to identify additional items not included in the initial list and allow for modifications in the wording of important items.[36]

## Study design and setting

The Delphi methodology is a widely used consensus-building technique whose main features are: use of experts as participants, anonymity between participants, iterations and controlled feedback (to allow for 'communication' and consensus building between participants), and summary of participant views.[40–43] Its key practical advantage is the non-requirement of face-to-face contact,[43] enabling more participants and broader representation in terms of geography and key groups in the development of health research reporting guidelines.[34] While our virtual Delphi approach provides participant anonymity that may allow for more open expression of views,[41] it has also the potential disadvantage of lack of group interaction which can contribute to consensus building.[43] However, the final consensus meeting (see below) will have an in-person element.

Our Delphi study will be conducted online and facilitated by DelphiManager software (V.5.0), a bespoke

software developed and maintained by the COMET (Core Outcome Measures in Effectiveness Trials) initiative (https://www.comet-initiative.org/), which has been used to develop other SPIRIT and CONSORT extensions.[44 45]

It is recommended that the number of Delphi study rounds is determined a priori.[40] To meet project timelines and reduce potential participant burden/fatigue, we propose two rounds will be conducted. Two Delphi rounds have been used to develop SPIRIT/CONSORT extensions for reporting: pilot RCTs,[46] RCTs using adaptive design,[35] interventions involving artificial intelligence,[44 47] and social and psychological interventions.[48] However, if we fail to obtain consensus in most items after round 2 and based on advice from the EC, we will consider a third round.[33 36 49]

### Sample size, recruitment and inclusion criteria

Delphi studies do not require formal sample size calculations, although the number and characteristics of participants need to be carefully considered.[40 42 43 50] Consistent with the EQUATOR Network's guidance,[34] we will seek to recruit an international multidisciplinary group of stakeholders, including trialists, trial methodologists, statisticians, healthcare professionals, researchers, content experts, journal editors, patient and public representatives, funders, regulators, health technology assessment experts and clinical guideline developers. Online supplemental file 1 shows stakeholder categories, approximate target sample sizes and strategies for access. We will target to have at least 200 participants register interest to take part in the study. A sample size of ≥200 will allow recruitment of at least 20 participants per stakeholder group and reasonable numbers (>70 participants) to complete the study, assuming ≥60% response rate of eligible participants and ≤40% attrition (in the context of completing all Delphi rounds), both as observed in previous Delphi studies that have developed extension guidelines.[35 36 46 47]

We will use purposive and snowball sampling (non-probability sampling) to include participants.[51] Identification strategies will include: (1) professional contacts known to the PMG and EC; (2) relevant professional bodies and networks; (3) relevant conferences and meetings; (4) authors of records included in the scoping and targeted reviews (phase 1); (5) a call for participants on project website and social media pages; (6) asking registered participants to share link with other people, networks or organisations that would be interested in participating (see online supplemental table) and (7) we have published short articles[52–54] to create awareness of the project and signpost readers to the project website to register interest in participation . We acknowledge that our target sample size may be challenging, particularly for some stakeholder categories (eg, journal editors (n=20)); however, efforts will be made to reach as many of these stakeholders as possible.

Participant inclusion criteria will be: (1) expertise in surrogate endpoints (through authored literature) or self-reported interest and basic understanding of the concept of surrogacy and (2) registered interest, in English (nevertheless participation will be international), to participate during the allocated period. We will have no exclusion criteria. During registration, the following descriptive data on participants will be collected: self-identified stakeholder group (primary/secondary roles), clinical or research area (if actively involved), country of work and self-reported basic understanding of surrogate endpoints.

### Data collection, analysis and consensus definition

Prior to the launch of the first survey round, a pilot (with n=15 participants) will be conducted to gauge user-friendliness, improve wording/logical flow and identify any practical concerns.[55] Following the pilot, all registered participants will be emailed a web link prompting them to complete any of the Delphi rounds. The weblink will access the Delphi survey landing page that includes: a short text section emphasising the importance of completing the exercise,[43] a participant information sheet and a consent checkbox. Each round will be open for approximately 4 weeks, and the second round sent out 3–4 weeks after the closure of the first. Email reminders to complete the survey will be sent to participants to improve response rates. Participants who complete all Delphi rounds may opt to enter a prize draw to win one of two £100 vouchers.[56]

Candidate Delphi items will be ranked using a Likert scale. There is no consensus on the ideal rating scale in Delphi studies.[57] We will use the Grading of Recommendations Assessment, Development and Evaluations (GRADE) 9-point scale[58] that has been used in other extensions,[33 35 44–46 56] categorised and interpreted as follows:

▶ 1–3=not important (item should not be included in extensions).
▶ 4–6=important but not critical (item should be discussed).
▶ 7–9=important and critical (item should be included in extensions).

Additionally, we will include an 'unable to answer' response, for participants who do not feel qualified to rank any specific item.[33 40] In round 1 of the Delphi,[33 56] participants will have the opportunity (through a free-text box) to add any proposed modification of items wording and suggestion of additional items.[36] We will consider new items in the second round if proposed by at least two Delphi participants.[59] Participants can also use the free-text box to explain any of their ratings.[36]

Quantitative analysis will be conducted in statistical software such as R.[60] After each round, the proportions for each candidate item will be calculated. In addition, measures to assess the consistency of response (agreement between Delphi rounds), such as median, IQR, mean and SD,[40] will be calculated for each item and round. Open text will be analysed using a simple thematic analysis[61] in Microsoft Excel sheets. Results after each round will be

shared with the project team before a virtual meeting to discuss modifications and additions needed for the subsequent round.[36]

Overall results and participants' own responses in round 1 will be shared with participants in round 2, and they can revise their judgement based on group scoring and explanation for ratings, if any. If conducted, the third round will only include items that have not reached a consensus by round 2.[56 62] Delphi studies have no ideal definition of consensus, and definitions vary with contexts.[40 57] We propose the following consensus definitions based on previous CONSORT extensions:[33 35 56 62]

► Consensus for inclusion: ≥70% participants scoring 7–9 and <15% participants scoring 1–3.
► Consensus for exclusion: ≥70% participants scoring 1–3 and <15% of participants scoring 7–9.
► No consensus for inclusion or exclusion: failure to achieve both the above.

Nevertheless, the project team will have the option to over-rule items that achieve a borderline level of consensus (eg, 65%–69% scoring 7–9).

### Phase 3: consensus meeting

The primary objective of the consensus meeting will be to agree on the final items for inclusion in the SPIRIT and CONSORT extensions. Dependent on time availability, this meeting will also be used as an opportunity to finalise and refine our knowledge translation strategies for the developed extensions.[34] The meeting will closely follow the EQUATOR Network's guidance on conducting face-to-face consensus meetings.[34]

#### Structure and participants

Given the ongoing COVID-19 pandemic and the range of geographical location of participants, we propose to conduct a hybrid meeting with some participants present physically and others joining virtually through an online video conference link. To allow sufficient time to discuss all agenda items[34] and participation from different time zones, we anticipate the meeting will be conducted over 2 days. Meeting participants will be limited to ≤30 to encourage interaction and debate.[34] Similar to the development of the CONSORT extension for adaptive design,[35] we propose the meeting will include the project team members (n=15) and purposively selected stakeholders (n=15). At the end of Delphi survey round 2, participants will be given the opportunity to register their interest in participating in the consensus meeting. The project team will select participants from those interested based on: (1) ability to attend the meeting on proposed dates and times and (2) need to have an international multidisciplinary group of participants.

Prior to the meeting, all participants will be sent (via email): the meeting agenda, participant list, meeting engagement rules, summary of scoping review findings and Delphi study results.[34] The meeting will be led by the EC chair (JSR).

#### Consensus procedure

Items that reached consensus during the Delphi exercise will be discussed and ratified. Items that did not reach consensus will be discussed in detail. The chair will summarise discussions and encourage consensus based on implications (ie, scientific, ethical, statistical, practical, financial) for inclusion or exclusion of discussed items before voting is conducted. Voting will be conducted anonymously with the following options: 'include in final extension', 'exclude from final extension', 'merge with another item' and 'unsure'.[33] Consensus will be defined as ≥70% voting for either the 'include', 'exclude' or 'merge' options.[33 35 45] Items that do not reach consensus will be discussed and a fresh round of voting conducted. This process will continue until consensus is reached or time allocated runs out.[33] If consensus is not reached at the end of the meeting, the PMG and EC will make a final decision[33] at a later virtual meeting held within a month after the consensus meeting. Discussion of items in both meetings will be audio recorded (with participant verbal consent) to help with comprehensive minuting (without direct reference to participants), which will provide a record of decisions taken.[34] Minutes will be shared with all participants after the meeting and archived with other project data.

After discussion of items and dependent on time availability, participants will discuss other agenda items such as the possibility of developing a flow diagram, knowledge translation activities (including authorship of extensions statements), strategies to improve implementation and impact of extensions, and future evaluation of developed outputs.[34]

### Phase 4: knowledge translation

This phase will aim to engage stakeholders and disseminate project outputs including the developed extensions and will be undertaken throughout the duration of project period.

#### Pilot testing and revision of final checklist

We will pilot the developed SPIRIT and CONSORT extension checklists[34] on a sample of protocols and reports identified from the targeted review. The pilot will involve project team members and other invited researchers and will seek to identify any specific challenges of using the draft checklists and required modifications and inform writing of the explanation and elaboration documents.

#### Publications

We will seek to publish the SPIRIT-SURROGATE and CONSORT-SURROGATE extensions and 'explanation and elaboration' documents in high impact general medicine journals. To maximise dissemination, we will also seek to co-publish the extensions or editorials/commentaries in other journal settings such as trial-related and public health. We will seek the endorsement of the extensions from journals and editorial groups (eg,

International Committee of Medical Journal Editors).[34] All project publications will be open access.

## Partner and stakeholder engagement

The EQUATOR, CONSORT and SPIRIT groups each provided letters of support for our original funding application for this project and have representatives of EQUATOR (GSC), SPIRIT (AWC) and international SPIRIT/CONSORT-Outcomes Group (NJB and MO) on our advisory EC. The project is registered on the EQUATOR website,[63 64] and our entry will be regularly updated. We will seek to have the final extensions endorsed and published on the SPIRIT[65] and CONSORT[66] websites. The SPIRIT statement is endorsed by patient groups, trial groups, funders, regulators and over 100 journals,[67] while the CONSORT statement is endorsed by about 600 journals.[68]

We will directly engage potential users of developed extensions, including research funders, healthcare regulators, trial methodologists, and public health and healthcare professionals, to maximise impact. Our project funding application was endorsed by the UK Medical and Healthcare products Regulatory Agency and the National Institute for Health and Care Excellence, who will be sent the finalised extensions for implementation. We will seek to reach other stakeholders through presentations at relevant meetings and conferences. We will maintain an active social media presence (ie, @Consort_surr on Twitter, project team members LinkedIn posts), a project ResearchGate page[69] and project website page (https://www.gla.ac.uk/spirit-consort-surrogate). Finally, we propose to develop video tutorials to illustrate the application of the extensions.

## Patient and public involvement

PPI will be embedded in all project phases. Our PPI strategy is led by DS who is a member of the PMG and has extensive experience of public involvement in health services research methods. The results of our scoping review will be presented and discussed with PPI representatives in a virtual meeting. We aim to identify ~20 PPI representatives (through snowballing guided by our PPI lead) who will be given introductory training (through a 2-hour session) on RCT design and the use of surrogate endpoints to build their capacity to participate in rating items in the Delphi survey. Some of the PPI members who complete the Delphi study will be selected to participate in the consensus meeting based on availability and interest to participate. We will consult our PPI lead on our public/community dissemination plans. The GRIPP2 (Guidance for Reporting Involvement of Patients and the Public) checklist[70] will be used to guide and report PPI.

## ETHICS AND DISSEMINATION

The study has received ethical approval from the University of Glasgow College of Medical, Veterinary and Life Sciences Ethics Committee (project no: 200210051).

Delphi participants will be provided with a participant information sheet and asked to record their online consent. Verbal consent for participation in the consensus meeting (including audio recording of discussions) will be sought at the start of the meeting. Participants will have a right to withdraw from any project activity without giving a reason. We anticipate that all project data collected will not be sensitive and that there is a low risk if the identity of participants was exposed. Nevertheless, all data will be securely managed and stored in a participant de-identified form. Delphi participants will be asked to opt-out (via a yes or no question) if they do not want to be publicly acknowledged in publications. Project outputs will be disseminated through meeting and conference presentations and open-access publications, with stakeholder engagement as outlined in the phase 4 (knowledge translation) and PPI sections.

**Author affiliations**
[1]MRC/CSO Social and Public Health Sciences Unit, School of Health and Wellbeing, Glasgow, UK, University of Glasgow, Glasgow, UK
[2]Population Health Sciences, Bristol Medical School, University of Bristol, Bristol, UK
[3]Patient and Public Involvement Lead, Nottingham, UK
[4]Edinburgh Clinical Trials Unit, Usher Institute, University of Edinburgh, Edinburgh, UK
[5]Department of Psychiatry, University of Toronto, Toronto, Ontario, Canada
[6]Child Health Evaluation Sciences, The Hospital for Sick Children, Toronto, Ontario, Canada
[7]Biostatistics Research Group, Department of Health Sciences, University of Leicester, Leicester, UK
[8]Women's College Institute Research Institute, Toronto, Ontario, Canada
[9]Department of Medicine, University of Toronto, Toronto, Ontario, Canada
[10]Centre for Statistics in Medicine, Nuffield Department of Orthopaedics, Rheumatology & Musculoskeletal Sciences, Oxford University, Oxford, UK
[11]National Institute for Health and Care Excellence, London, UK
[12]AstraZeneca, Mölndal, Sweden
[13]Department of Health Policy and Management, Yale School of Public Health, New Haven, Connecticut, USA
[14]Section of General Medicine, Department of Internal Medicine, Yale School of Medicine, New Haven, Connecticut, USA
[15]Robertson Centre for Biostatistics, School of Health and Well Being, University of Glasgow, Glasgow, UK
[16]SDA Bocconi School of Management, Milan, Italy

**Acknowledgements** We wish to acknowledge our dear colleague Professor Amber Young who lost her brave battle with cancer and passed away on 17th September 2022. The authors would like to thank Valerie Wells (University of Glasgow) for helping with development of a search strategy for the literature reviews and Rui Duarte and Rebecca Bresnahan (both from University of Liverpool) for their advice during conceptualisation of Delphi study design.

**Contributors** PD, CW, AY, RT and OC were involved in funding acquisition. AMM, PD, DS, CW, AY, RT and OC were involved in the initial phases of project phases conception and design. AMM, RT and OC were responsible for the first draft of the manuscript. NJB, SB, AWC, GSC, DD, MO, Mario Ouwens and JSR critically reviewed the first draft and approved the final version. The views expressed in this article are those of the authors and not their employers or funders.

**Funding** The development of SPIRIT and CONSORT extensions has been funded by the UK Medical Research Council (grant number MR/V038400/1). GSC was supported by the NIHR Biomedical Research Centre, Oxford, and Cancer Research UK (programme grant: C49297/A27294).

**Competing interests** Sylwia Bujkiewicz has served as a paid consultant, providing methodological advice, to NICE, Roche and RTI Health Solutions, received payments for educational events from Roche and has received research funding from

European Federation of Pharmaceutical Industries and Associations (EEPIA) and Johnson and Johnson. Mario Ouwens works for and has shares in AstraZeneca. Dr Joseph S Ross is an Associate Editor at BMJ and co-founder (unpaid) of medRxiv; Dr Ross currently receives research support through Yale University from Johnson and Johnson to develop methods of clinical trial data sharing, from the Medical Device Innovation Consortium as part of the National Evaluation System for Health Technology (NEST), from the Food and Drug Administration for the Yale-Mayo Clinic Centre for Excellence in Regulatory Science and Innovation (CERSI) programme (U01FD005938), from the Agency for Healthcare Research and Quality (R01HS022882), from the National Heart, Lung and Blood Institute of the National Institutes of Health (NIH) (R01HS025164 and R01HL144644), and from the Laura and John Arnold Foundation to establish the Good Pharma Scorecard at Bioethics International; in addition, Dr Ross is an expert witness at the request of Relator's attorneys, the Greene Law Firm, in a qui tam suit alleging violations of the False Claims Act and Anti-Kickback Statute against Biogen Inc.

**Patient and public involvement** Patients and/or the public were involved in the design, or conduct, or reporting, or dissemination plans of this research. Refer to the Methods section for further details.

**Patient consent for publication** Not applicable.

**Provenance and peer review** Not commissioned; externally peer reviewed.

**ORCID iDs**
Anthony Muchai Manyara http://orcid.org/0000-0001-6276-926X
Christopher J Weir http://orcid.org/0000-0002-6494-4903
Amber Young http://orcid.org/0000-0001-7205-492X
Nancy J Butcher http://orcid.org/0000-0002-5152-0108
Gary S Collins http://orcid.org/0000-0002-2772-2316
Martin Offringa http://orcid.org/0000-0002-4402-5299

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
