## [Reviewer comments · BMJ Open]

ARTICLE DETAILS

TITLE (PROVISIONAL)	Protocol for the development of SPIRIT and CONSORT extensions for randomised controlled trials with surrogate primary endpoints: SPIRIT-SURROGATE and CONSORT-SURROGATE.
AUTHORS	Manyara, Anthony; Davies, Philippa; Stewart, Derek; Weir, Christopher; Young, Amber; Butcher, Nancy; Bujkiewicz, Sylwia; Chan, An-Wen; Collins, Gary; Dawoud, Dalia; Offringa, Martin; Ouwens, Mario; Ross, Joseph; Taylor, Rod; Ciani, Oriana

VERSION 1 – REVIEW

REVIEWER	Saint-Hilary, Gaelle Politecnico di Torino, Department of Mathematics
REVIEW RETURNED	16-Jun-2022

GENERAL COMMENTS	Referee report on BMJOpen-2022-064304: “Protocol for the development of SPIRIT and CONSORT extensions for randomized controlled trials with surrogate endpoints: SPIRIT-SURROGATE and CONSORT-SURROGATE” The manuscript is a protocol for the development of guidelines for clinical study protocols (extension of the SPIRIT guideline) and clinical study reports (extension of the CONSORT guideline) for clinical trials using surrogate endpoints. Early endpoints, i.e. markers considered to be predictive of the treatment effect on clinical endpoints of interest, are widely used in clinical trials. However, they are not necessarily all accepted as surrogates from a regulatory perspective, and the evidence regarding the relationship between these so-called surrogate endpoints and the clinical endpoints of interest may be limited. It is therefore relevant to propose some guidance to present and report trials using surrogate endpoints, with a scientific rationale for their choice, to ensure that the objectives of these clinical trials are attainable and that their results are interpretable. Therefore, the purpose of the proposed protocol is relevant. The proposed methodology is clear, well-structured, and similar to what was successfully used for the development of previous SPIRIT and CONSORT extensions. The list of stakeholders to be involved in their development is relevant.
--

	Overall, the proposed protocol is therefore suitable for publication in BMJ Open. Below are some suggestions for improvement of the protocol or the process. Introduction, page 5, lines 14-18: These are the advantages for the sponsor, but using (good) surrogate endpoints is also beneficial to the patients. It permits: To detect lack of efficacy early, allowing to stop a trial / drug development early and then avoiding that many patients receive a non-effective treatment; To detect efficacy early, permitting in some cases to obtain a conditional approval (provided that the surrogate is accepted by the regulatory authorities) so that the whole population of patients can benefit from the efficacious treatment earlier. Introduction, page 6, lines 13-14: one other advantage of these guidelines would be to help the sponsors understand / realize that some evidence is lacking to use the endpoints they have identified as surrogate. When referring to other SPIRIT and CONSORT extensions, you could also refer to the on-going DEFINE study for the development of SPIRIT and CONSORT extensions for dose-finding clinical studies, using the same methodology, https://www.icr.ac.uk/DFCONSORT. Phase 2 Delphi study, page 7, lines 56-57: avoiding face-to-face contact in elicitation processes has also some limitations that should be highlighted too. Recruitment, page 8, lines 10-34: although statisticians are certainly included in the methodologists, I suggest to mention them explicitly since they are those who can scientifically prove the surrogacy of an endpoint. Partner and stakeholder engagement, page 13, lines 7-16: that would be important to have endorsement from regulatory bodies outside of UK, and to get their engagement before finalizing the protocol. Although the objective of this protocol is valuable, the authors should keep in mind that the multiplication of guidelines also increases the complexity of drug developments – complexity that is actually acknowledged in introduction. The authors are then encouraged to: Make the new guidelines as short and simple as possible. Highlight clearly the differences / additions between these extensions and the original SPIRIT and CONSORT guidelines, so that the sponsors can rapidly ensure their trial complies with both the original versions and the extensions. Be careful that the proposed guidelines do not prevent the use of exploratory markers not yet proven to be surrogates, since this proof necessitates data to be collected at some point. Not propose these guideline extensions in the end, if it appears that the differences between the original versions and the extensions are minimal, or their added value limited. That should not be considered as a failure, since the whole process that would lead to that conclusion would be of interest (and should be published).
--	---

REVIEWER	Molenberghs, Geert Universiteit Hasselt, Interuniversity Institute for Biostatistics and statistical Bioinformatics (I-BioStat)
REVIEW RETURNED	26-Jun-2022

GENERAL COMMENTS	This is a very valuable project that has the potential to contribute in important ways to not only the use but also the proper use of surrogate marker methodology in clinical trials, and the reporting thereof. In surrogate marker evaluation, one distinguishes between two levels:  • Individual-level surrogacy, addressing the question how well the surrogate is predictive for the true endpoint at the level of the individual patient, given the treatment arm and perhaps other covariates. • Trial-level surrogacy, addressing the question how well the treatment effect on the surrogate is predictive for the treatment effect on the true endpoint. This is typically evaluated across a number of trials (meta-analytic framework) or centers. There are several evaluation paradigms: apart from the already mentioned meta-analytic framework, there is the principal-stratification framework, the methodology based on information theory, and various causal-inference based methods. I find that the above is largely absent from the paper/protocol, which could be seen as a noticeable omission. It is, of course, fine that a study protocol does not address each and every aspect, but a clear survey of the land and then delineating of the study's objectives would enhance clarity and align the reader's expectations. When a study is evaluated, the above should be part of it – i.e., does it allow to evaluate/validate surrogacy at the individual level and/or at the trial level. References 9 and 10 are important towards understanding that surrogates have generated adverse results, to say the least. Of course, these are somewhat older and there is an entire literature available on the entire research field. Generally, the paper could include some more methodological, i.e., statistical, references. Cf. also the book references, e.g., Alonso A., Bigirumurame T, Burzykowski T, Buyse M, Molenberghs G, Muchene L, Perualila NJ, Shkedy Z, Van der Elst W (2017). Applied Surrogate Endpoint Evaluation with SAS and R. Boca Raton: Chapman&Hall/CRC. I can see the rationale for snowball sampling, but there are important bias concerns, of course.
--

VERSION 1 – AUTHOR RESPONSE

Reviewer 1		
The manuscript is a protocol for the development of guidelines for clinical study protocols (extension of the SPIRIT guideline) and clinical study reports (extension of the CONSORT guideline) for clinical trials using surrogate endpoints. Early endpoints, i.e., markers considered to be predictive of the treatment effect on clinical endpoints of interest, are widely used in clinical trials. However, they are not necessarily all accepted as surrogates from a regulatory perspective, and the evidence regarding the relationship between these so-called surrogate endpoints and the clinical endpoints of interest may be	Thank you	No change.

limited. It is therefore relevant to propose some guidance to present and report trials using surrogate endpoints, with a scientific rationale for their choice, to ensure that the objectives of these clinical trials are attainable and that their results are interpretable. Therefore, the purpose of the proposed protocol is relevant. The proposed methodology is clear, well-structured, and similar to what was successfully used for the development of previous SPIRIT and CONSORT extensions. The list of stakeholders to be involved in their development is relevant.		
Overall, the proposed protocol is therefore suitable for publication in BMJ Open. Below are some suggestions for improvement of the protocol or the process.  1. Introduction, page 5, lines 14-18: These are the advantages for the sponsor but using (good) surrogate endpoints is also beneficial to the patients. It permits:  a. To detect lack of efficacy early, allowing to stop a trial / drug development early and then avoiding that many patients receive a non-effective treatment. b. To detect efficacy early, permitting in some cases to obtain a conditional approval (provided that the surrogate is accepted by the regulatory authorities) so that the whole population of patients can benefit from the efficacious treatment earlier. 	Thank you for these suggestions. We have added statements to capture both benefits and in addition detection of safety.	Addition of the following statements on the first paragraph of the introduction, page 5: This efficiency allows for early detection of intervention effects⁶ which could lead to accelerated approval of interventions prior to confirmation of benefit on the PRFO⁷ or when there is lack of effect efficacy, stopping of trials or roll out of interventions with no health benefit.
 2. Introduction, page 6, lines 13-14: one other advantage of these guidelines would be to help the sponsors understand / realize that some evidence is lacking to use the endpoints they have identified as surrogate. 	We have now included statements to capture that guideline would contribute to better scrutiny of surrogacy evidence	The following statements have been added on page 6, last paragraph of the introduction: These extensions will improve the transparency of reporting and design of RCTs with a surrogate endpoint primary outcome. Such improved

		reporting should enable the evidence base for surrogate endpoints to be more effectively scrutinised and used for interpretation of trial findings by patients, clinicians, and healthcare policy makers. This protocol describes the methods that will be used in developing these extensions.
3. When referring to other SPIRIT and CONSORT extensions, you could also refer to the on-going DEFINE study for the development of SPIRIT and CONSORT extensions for dose-finding clinical studies, using the same methodology, https://www.icr.ac.uk/DFCONSORT.	Thank you for this addition, we have now included it in the methodological considerations we are drawing from.	The first paragraph of the Methodology on page 6 now reads: Our methodology will be guided by: the EQUATOR (Enhancing the QUALity and Transparency Of health Research) Network's recommended steps for developing a health research reporting guideline 34; and methodological considerations used to develop other recent or ongoing SPIRIT/CONSORT extensions (e.g., ACE [Adaptive designs CONSORT Extension] 35,

		CONSORT-ROUTINE 36, Dose-Finding CONSORT Extension 37).
4. Phase 2 Delphi study, page 7, lines 56-57: avoiding face-to-face contact in elicitation processes has also some limitations that should be highlighted too.	We have added a statement to capture the strength but also limitation of lack of face-to-face contact	Addition of the statement below on the first paragraph on page 8: Whilst our virtual Delphi approach provides participant anonymity that may allow for more open expression of views ⁴¹ , it has also the potential disadvantage of lack of group interaction which can contribute to consensus building ⁴³ . However, the final consensus meeting (see below) will have an in-person element.
5. Recruitment, page 8, lines 10-34: although statisticians are certainly included in the methodologists, I suggest mentioning them explicitly since they are those who can scientifically prove the surrogacy of an endpoint.	Thank you for this suggestion.	Addition of "statisticians" in mentioned stakeholders on page 9 and supplementary table 1
6. Partner and stakeholder engagement, page 13, lines 7-16: that would be important to have endorsement from regulatory bodies outside of UK, and to get their engagement before finalizing the protocol.	Thank you. We are targeting regulators as one of the stakeholder groups to participate in our Delphi survey and consequently in the consensus meeting.	No change
7. Although the objective of this protocol is valuable, the authors should keep in mind that the multiplication of guidelines also increases	Thank you for these suggestions, we will reflect on	No change

the complexity of drug developments – complexity that is actually acknowledged in introduction. The authors are then encouraged to:  a. Make the new guidelines as short and simple as possible. b. Highlight clearly the differences / additions between these extensions and the original SPIRIT and CONSORT guidelines, so that the sponsors can rapidly ensure their trial complies with both the original versions and the extensions. c. Be careful that the proposed guidelines do not prevent the use of exploratory markers not yet proven to be surrogates, since this proof necessitates data to be collected at some point. d. Not propose these guideline extensions in the end, if it appears that the differences between the original versions and the extensions are minimal, or their added value limited. That should not be considered as a failure, since the whole process that would lead to that conclusion would be of interest (and should be published). 	them throughout the development process of the guidelines and when writing the explanation and elaboration document along with the checklist	
Reviewer 2		
This is a very valuable project that has the potential to contribute in important ways to not only the use but also the proper use of surrogate marker methodology in clinical trials, and the reporting thereof.	Thank you.	No change.
In surrogate marker evaluation, one distinguishes between two levels:  • Individual-level surrogacy, addressing the question how well the surrogate is predictive for the true endpoint at the level of the individual patient, given the treatment arm and perhaps other covariates. • Trial-level surrogacy, addressing the question how well the treatment effect on the surrogate is predictive for the treatment effect on the true endpoint. This is typically evaluated across a number of trials (meta-analytic framework) or centres. There are several evaluation paradigms: apart from the already mentioned meta-analytic framework, there is the principal-stratification framework, the methodology based on information theory, and various causal-inference based methods. 		

VERSION 2 – REVIEW

REVIEWER	Saint-Hilary, Gaelle Politecnico di Torino, Department of Mathematics
REVIEW RETURNED	18-Sep-2022

GENERAL COMMENTS	My comments have been addressed, I think the protocol is good for publication.
--

REVIEWER	Molenberghs, Geert Universiteit Hasselt, Interuniversity Institute for Biostatistics and statistical Bioinformatics (I-BioStat)
REVIEW RETURNED	14-Aug-2022

GENERAL COMMENTS	The authors have adequately addressed the points that I raised. I just have a few small points: * In response to Reviewer 1, point 1, the additional wording is fine, but I think that the wording can be improved a bit. * In response to Reviewer 2, the point addressed on page 4 of the 'Replies' document, I would suggest to replace "and bivariate network..." by "and/or bivariate network...". Indeed, while the methods listed are appropriate, it is not necessary to apply all of them, but this may be what readers understand otherwise.
---